# Enhanced emergent electromagnetic inductance in Tb$_5$Sb$_3$ due to highly disordered helimagnetism
Aki Kitaori [1,2,3] ✉, Jonathan S. White [4], Victor Ukleev [4,5], Licong Peng [3], Kiyomi Nakajima[3], Naoya Kanazawa [6] ✉, Xiuzhen Yu [3], Yoshichika Ōnuki[3] & Yoshinori Tokura [2,3,7] ✉

In helimagnetic metals, ac current-driven spin motions can generate emergent electric fields acting on conduction electrons, leading to emergent electromagnetic induction (EEMI). Recent experiments reveal the EEMI signal generally shows a strongly current-nonlinear response. In this study, we investigate the EEMI of Tb$_5$Sb$_3$, a short-period helimagnet. Using small angle neutron scattering we show that Tb$_5$Sb$_3$ hosts highly disordered helimagnetism with a distribution of spin-helix periodicity. The current-nonlinear dynamics of the disordered spin helix in Tb$_5$Sb$_3$ indeed shows up as the nonlinear electrical resistivity (real part of ac resistivity), and even more clearly as a nonlinear and huge EEMI (imaginary part of ac resistivity) response. The magnitude of the EEMI reaches as large as several tens of µH for Tb5Sb3 devices on the scale of several tens of µm, originating to noncollinear spin textures possibly even without long-range helimagnetic order.

Inductors are one of the major components of analog electrical circuits, which produce a phase difference in alternating currents and voltages by Faraday's law of electromagnetic induction. The performance of inductors is characterized by the relation $V = LdI/dt$, where $V$, $I$, and $L$ are voltage, current and inductance, respectively. With conventional inductors based on classical electromagnetism, inductance is proportional to $n^2A$, $n$ and $A$ being the number of coil windings and the coil cross-section, respectively. Thus, it would be practically difficult to dramatically reduce the dimension of the inductor of coil form whilst retaining large inductances. Recently, a new concept of inductor element has been proposed that utilizes the current-driven motion of spin textures[1]. This is termed emergent electromagnetic inductance (EEMI), since it is based on the emergent electric field (**e**) generated via the dynamics of chiral spin textures[2,3]. The roots of the EEMI lie with the emergent magnetic field (**b**) acting on conduction electrons, which is realized for non-coplanar spin textures endowed with scalar spin chirality[4–7]. The generalized Faraday's law[8] tells that $\nabla \times \mathbf{e} = -\partial \mathbf{b}/\partial t$ or $\mathbf{e} = -\partial \mathbf{a}/\partial t$, where **a** is the Berry connection satisfying $\mathbf{b} = \nabla \times \mathbf{a}$. Here, an originally static **b** is not necessarily required for generating non-zero **e**, since it can be generated according to a time dependent deformation of the spiral magnetic structure. The coordinate component of emergent electric field ($e_i$) is described as[9];

$$e_i = \frac{h}{2\pi e}\mathbf{n} \bullet \left(\partial_i \mathbf{n} \times \partial_t \mathbf{n}\right),$$

where **n**, $h$ and $e$ are a unit vector parallel to the direction of spin moment, Planck's constant, and bare electron charge, respectively. As opposed to **b**, **e** is related to the dynamics of spin structures and proportional to the solid angle dynamically swept by $\mathbf{n}(t)$. Such an effect has been discussed for the domain wall motion and related systems[10]. In particular, for the case of a proper-screw helix[1,11,12], the emergent electric field[9–16] can be described as[1],

$$e_x = \frac{Ph}{e\lambda}\partial_t\phi,$$

where $\lambda$ is the period of helix, $P$ is a spin polarization factor, $\varphi$ is the tilting angle of the spin from the spiral plane, and the $x$-axis is taken parallel to the magnetic modulation vector (**q**). To obtain a large emergent field, the short period $\lambda$, $e.g.$, a few nanometers, is essential, which often originates inside

[1]Institute of Engineering Innovation, The University of Tokyo, Tokyo 113-0032, Japan. [2]Department of Applied Physics, The University of Tokyo, Tokyo 113-8656, Japan. [3]RIKEN Center for Emergent Matter Science (CEMS), Wako 351-0198, Japan. [4]Laboratory for Neutron Scattering and Imaging (LNS), Paul Scherrer Institute (PSI), CH-5232 Villigen, Switzerland. [5]Helmholtz-Zentrum Berlin für Materialien und Energie, D-14109 Berlin, Germany. [6]Institute of Industrial Science, The University of Tokyo, Tokyo 153-8505, Japan. [7]Tokyo College, The University of Tokyo, Tokyo 113-8656, Japan. ✉e-mail: kitaori@ap.t.u-tokyo.ac.jp; naoya-k@iis.u-tokyo.ac.jp; tokura@riken.jp

magnetic materials due to frustration effects. Given that the tilting motion of spins follows the applied ac current, 90° phase difference exists between the emergent electric field and the current. Furthermore, higher current frequencies lead larger emergent electric field because it originates from the time-derivative term of spin motion. The emergent electric field works like the electric field caused by the inductance $L$.

This is the EEMI based on the tilting mode, which is the simplest mechanism of emergent inductance generation. Contrary to classical inductors, an inductor based on EEMI has an inductance value that is inversely proportional to the cross-sectional area of the device. In addition, the structure is simple, consisting of only a helimagnet and electrodes. Therefore, the EEMI effect in helimagnets contributes to a dramatic miniaturization of inductance elements. There are several reports not only on theoretical prediction[17–19] but also on specific experimental examples[20,21]. Among them, room-temperature inductors using $YMn_6Sn_6$ show inductance values comparable with of those of commercial products, but with the volume of the inductor element reduced by a factor of about 100,000.

So far, several non-trivial experimental features have been clarified through studies on EEMI. Analyzing the spin motion that produces the emergent inductance, the contribution of the nonlinear behavior with respect the applied current is large and often dominating the observed results[20–22]. Specifically for helically-ordered magnetic phases, ac current induced motion has not only the tilting mode component, but also the spin-helix translational mode (or the phason mode) component. With the latter mode, spin rotates uniformly within the plane while propagating along the **q** direction as shown in Fig. 1a[21,22]. In particular, the pinned motion of the phason under the ac-current excitation is expected to cause the negative EEMI[15]. In fact, a large negative EEMI response is observed for long-range ordered helimagnetic textures, such as spin screw and transverse conical states, in the relatively low ac-current frequency and low temperature regimes and is ascribed to the pinned phason dynamics[20–22].

It is to be noted that the positive EEMI response is also observed upon ac current excitation of the helical or even the spin-collinear antiferromagnetic state close to the magnetic phase transition temperatures. Here the thermal fluctuation of the constituent spins, and hence thermally-induced spin noncollinearity are anticipated to be the cause[21,22]. Since the origin of the emergent electric field generation is due to the temporal deformation of noncollinear neighbouring spin pairs, the exploration of highly disordered spin ensembles, such as disordered short-period spin helices and cluster spin-glasses that couple with the charge dynamics, also become important. In such disordered non-collinear magnetic systems, the current-nonlinear dynamics may be manifested by the nonlinear resistivity, which would also lead to a highly current-nonlinear EEMI. However, the impacts of disorder in spin helices on their current-induced dynamics and EEMI therein remain elusive. To explore such possibly new aspects of EEMI, we target here the intermetallic compound $Tb_5Sb_3$ with short-period and highly disordered helimagnetic spin configurations.

## Results and discussion
### Magnetic structure

$R_5M_3$ ($R$ = La-Nd, Gd-Lu, and $M$ = Ge, Sn, Pb, and Sb) compounds crystallize in the $Mn_5Si_3$-type structure ($P6_3/mcm$)[23]. As shown in Fig. 1b, there are two inequivalent crystallographic sites of magnetic R ion. Due to the magnetic frustration between these sites, $R_5M_3$ group magnets host various short-period spiral spin textures[24–26]. Some members of this family, such as $Nd_5Ge_3$[27,28], are known to exhibit glassy spin cluster behavior. $Tb_5Sb_3$ too is known for its enigmatic magnetism[29–33]. This material undergoes two broad magnetic transitions that manifest clearly in both magnetization and specific heat. Figure 1c shows the temperature dependence of magnetization, which reveals a lower transition temperature $T_{C1}$ (~55 K) and a higher transition temperature $T_{C2}$ (~133 K). From neutron scattering measurements on the polycrystalline specimens reported by two independent research groups[29,30], the magnetic phase for $T_{C1} < T < T_{C2}$ is assigned to be the **q** // $c$-axis conical phase with tilted cone. On the other hand, for the magnetic phase with $T < T_{C1}$, both studies confirmed the existence of the in-

plane propagation vector **q** $\perp c$, but the detailed magnetic structures remained elusive. A particular difficulty in determining the magnetic structures of this material, especially for the lower-temperature ($T < T_{C1}$) phase, has been a lack in the availability of single crystalline samples.

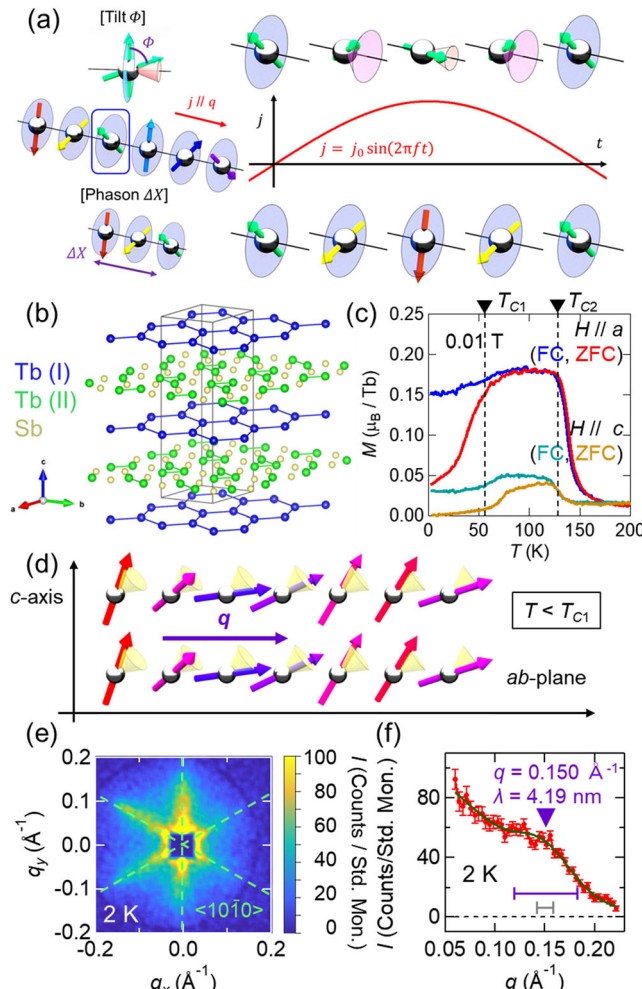

**Fig. 1 | Crystal and magnetic structure of $Tb_5Sb_3$. a** Schematic illustrations of spin-tilting mode and phason mode of the spin-helix state responsible for emergent electromagnetic induction during the half cycle of the ac current $j = j_0 \sin(2\pi ft)$ excitation. **b** Crystal structure of $Tb_5Sb_3$. For clarity, the $c$-axis direction is shown with a length that is 5 times longer. There are two types of Tb sites: Tb(I) (blue) forming a honeycomb lattice and Tb(II) (green) triangle trimer forming a lattice with Sb. The illustration was drawn using VESTA[37]. **c** Temperature dependence of magnetization. Blue, red, green, and yellow curves correspond to the cases of $H$ // $a$ (field cooling), $H$ // $a$ (zero field cooling), $H$ // $c$ (field cooling), $H$ // $c$ (zero field cooling). Two transition temperatures ($T_{C1}$, $T_{C2}$) are indicated by black dashed lines. **d** Illustration of the tilted conical phase, the basic unit of the magnetic structure at $T < T_{C1}$. **e** SANS pattern of $Tb_5Sb_3$ at 2 K, 0 T. Streak-like signals appear along the six directions equivalent to [10-10] direction (pale green dashed lines). An absorption ring-like feature seen in the outer edge of the detector arises due to absorption of the outgoing beam by part of the SANS detector construction. The affected part of the detector is neglected in the SANS data analysis. **f** $q$ dependence of the SANS intensity. The red symbols and green curve respectively correspond to the measurement data and the fitting curve. The full width at half-maximum (FWHM) at each point is shown as an error bar. The latter is fitted by a combination of two gaussian functions, one of which describes ferrimagnetic-like correlations centered at $q = 0$, with the second gaussian describing the helical modulation at centered at finite $q$ (see the Supplementary Note 1 for full details). For the latter, the peak center position and FWHM are indicated with purple triangle and horizontal bar, respectively. The gray horizontal bar indicates the instrumental resolution, which would correspond to the FWHM of a sharp peak expected for the typical case for long-range incommensurate order centered at the same $q$.

Recently, we have succeeded in single crystal growth of this incongruent-melting material by the self-flux method as described elsewhere[33]. Fig. 1c shows the temperature dependent magnetization measured using a single crystal sample. Both $M$-$T$ curves with $H \parallel c$ and $H \perp c$ show spontaneous magnetization in the region $T < T_{C1}$. Considering the magnetic modulation in the $ab$-plane around 4 nm, which is a consensus of previous studies[29,30], the magnetic structure at the lowest temperature can be assigned to a derivative of a tilted conical phase in which the **q** vector propagates in the $ab$-plane as illustrated in Fig. 1d.

The magnetic features below $T_{C1}$ were furthermore investigated by small angle neutron scattering (SANS) measurements on a $Tb_5Sb_3$ single crystal (see Methods for further experimental details). By collecting detailed SANS data, we revealed the short-period spiral magnetism in this material to be highly disordered. Figure 1e shows the SANS pattern at $T = 2$ K and zero magnetic field. In case of conventional helimagnets, the SANS pattern displays sharp spots at locations in reciprocal space that correspond to the length and direction of the real-space modulation. In $Tb_5Sb_3$, however, the SANS intensity forms a streak pattern, with the streaks extending from the origin, **q** = (0 0 0). Careful analysis of the streaks reveals them to be structured, and we interpret them to be due to a combination of possible ferrimagnetic domain correlations, and a broad peak corresponding to a spiral spin texture. The broad peak appears at $q = 0.150$ Å$^{-1}$ ($\lambda = 4.12$ nm) in the $q$ dependence of the SANS intensity [Fig. 1f] (see the Supplementary Note 1 for full detail of the SANS data analysis). The peak center corresponds well to the size of the primitive unit of magnetic modulation (~4 nm) observed in previous powder neutron diffraction measurements. The full-width at half maximum (FWHM) of the peak is as large as 0.032 Å$^{-1}$, which is clearly broader than the instrument resolution (0.016 Å$^{-1}$), and thus evidences the formation of disordered helical textures with an estimated in-plane correlation length of only ~1 nm, or just a single in-plane lattice constant. With only the sixfold-symmetric SANS intensity distribution, however, it is difficult to distinguish between skyrmion-like phase (multiple-$q$, $i.e.$, 3-$q$ state) and single-$q$ helical magnetic order (1-$q$ state). From the results of Lorentz transmission electron microscopy (LTEM) also performed on this material (see also the Supplementary Note 2), it is likely that the main phase in the bulk is the single-$q$ helical magnetic order (Fig. 1d).

The magnetic structure of $Tb_5Sb_3$ varies not only with respect to the $q$ value but also with respect to the in-plane $q$-orientation. Figure 2 shows SANS patterns taken at various temperatures. At low temperatures, $e.g.$, $T = 10$ K, the $q$ vectors propagate along the [10-10] and equivalent

directions, like the case of $T = 2$ K. As the temperature increases, streaks also appear in the [11-20] and equivalent directions, which are rotated by 30° from [10-10] direction. When the temperature is increased further up to near the transition temperature $T_{C1}$ ( = 55 K), the whole pattern disappears. Since it is rare for the direction of the $q$ vector to change with temperature in this way, our observations indicate that the orientation of the $q$ vector is not strongly fixed to a specific crystallographic axis in the $ab$-plane, and hence that the spin structure may be easily driven by external stimulations such as electric-current and thermal flows.

## Nonlinear resistivity and emergent inductance

Keeping in mind the existence of disordered and flexible magnetic textures below $T_{C1}$, we now turn to the electric transport phenomena on $Tb_5Sb_3$. Before discussing the emergent inductance, we first show the real part of ac electrical resistivity (Re $\rho$) with the ac current excitations. Occasionally, a bulk $Tb_5Sb_3$ crystal ingot contains coexisting microcrystals of TbSb as an impurity phase. Since TbSb is a good metal that shows very low resistivity (~0.18 μΩ cm at $T = 2$ K) at low temperature[34,35], it is difficult to precisely measure electrical resistivity on a bulk $Tb_5Sb_3$ crystal without the possible influence of TbSb forming parallel conduction paths. Here, we attempted to perform accurate charge transport measurements of $Tb_5Sb_3$ by cutting out a

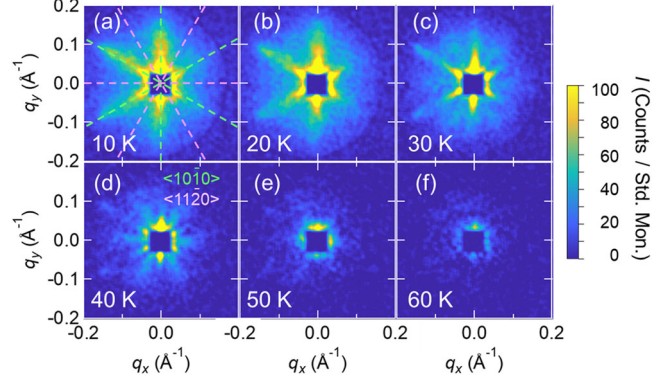

**Fig. 2 | The direction of $q$ vectors in $Tb_5Sb_3$. a–f** Temperature dependence of the SANS pattern from 10 K to 60 K. Pale green dashed lines show the direction equivalent to [10-10] and pale pink dashed lines show the direction equivalent to [11-20] in $ab$-plane.

**Fig. 3 | Nonlinear behavior of resistivity.** Temperature dependence of the **(a)** real part and **(b)** imaginary part of ac resistivity with various current densities and frequencies. The color in the curves in **(a)** and **(b)** represents individual current density $j_0$, while the shape of the markers in **(a)** distinguish frequency $f$. In the present device size, Im $\rho = 10$ μΩ cm corresponds to $L = 14$ μH. Transition temperatures ($T_{C1}$, $T_{C2}$) are determined from magnetization curve as shown in Fig. 1b.

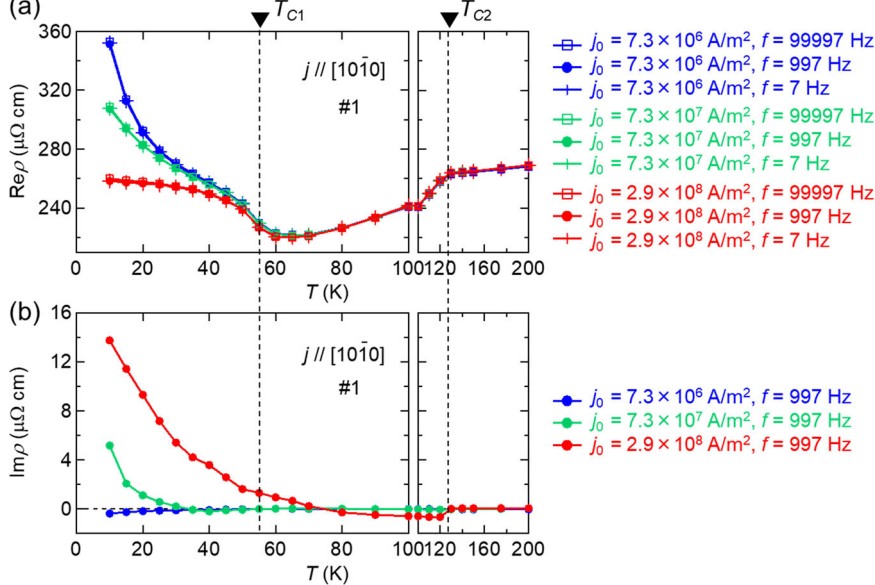

**Fig. 4 | Emergent inductance. a** Frequency dependence of the emergent inductance at $T = 10$ K with various current densities. **b** Current density dependence of the inductance at $f = 105, 1000, 2000$ Hz. **c** Frequency dependence of the inductance at various temperatures from $T = 10$ K to 60 K and at a relatively weak current excitation, $j_0 = 7.3 \times 10^7$ A/m$^2$.

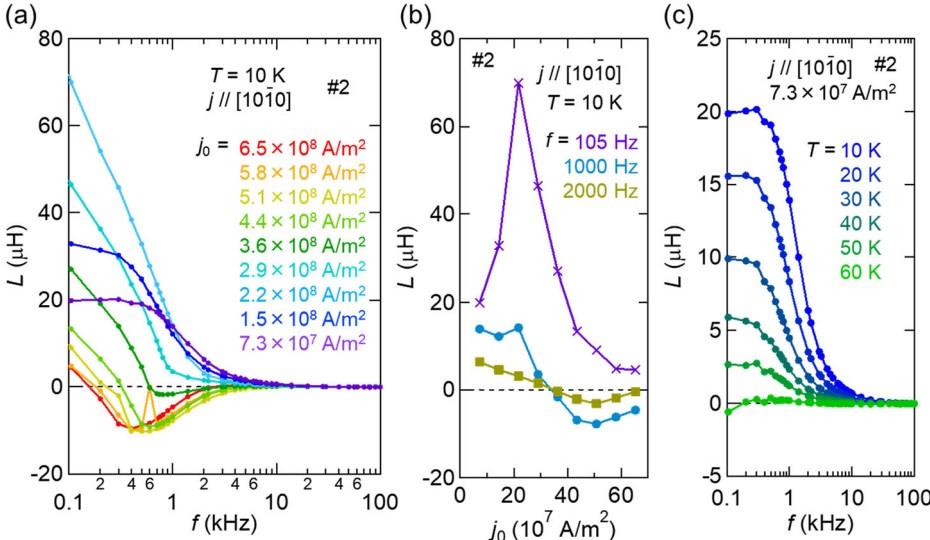

small region of genuine Tb$_5$Sb$_3$ from a bulk crystalline sample by using the focused ion beam (FIB) technique. Figure 3a shows the temperature dependence of electrical resistivity (real part) measured with a lock-in amplifier, using the ac input current density $j = j_0 \sin(2\pi f t)$; $j_0 = 0.73 - 6.5 \times 10^8$ A/m$^2$ and $f$ is frequency (see the Supplementary Note 3 for magnetic field dependence of transport). Here the device (#1) size is 3.2 μm × 8.6 μm in cross section ($S$) and 24.5 μm long in voltage-terminal distance ($d$).

Below the magnetic transition temperature $T_{C1}$ (= 55 K) to the disordered helix phase, the Re $\rho$ tends to increase with lowering temperature like a semiconductor, and also strongly depends on the current density, while essentially no current-density dependence of Re $\rho$ is discerned above $T_{C1}$ in the present current-density regime of $j_0 = 0.73 - 6.5 \times 10^8$ A/m$^2$. As a possible source of the strong nonlinearity in resistivity, the current-induced heating effect of the sample can be excluded, since the resistivity value at any fixed temperature shows almost no current-frequency dependence over a wide $f$-range of 7 Hz-10 kHz. Thus, the onset of nonlinear resistivity present only below $T_{C1}$ indicates that the low-temperature magnetic structure, i.e., the disordered helix state, couples with the charge transport in a $j$-dependent manner. A rather insulating temperature dependence of the low-current resistivity is reminiscent of a charge-gap opening or charge-density-wave (CDW) formation associated with short-period spin helix order of Ruderman-Kittel-Kasuya-Yosida (RKKY) interaction origin. The observed suppression of the resistivity with increasing current-density may correspond to a current-induced depinning of such a spin/charge density wave[36]. Although the highly disordered helical magnetism, as well as the lack in clean sufficient-size single crystals of Tb$_5$Sn$_3$, hinders the detailed characterization of the electronic structure below $T_{C1}$ at the moment, the helical spin modulation accompanying a possible CDW-like gap is an important issue worthy of further study. This spin-texture coupled charge transport as well as its strong nonlinear effect arising from current-induced spin-texture dynamics is likely to be closely related to the EEMI characteristics as investigated in the following.

Figure 3b shows the temperature dependence of the imaginary part of the electrical resistivity (Im $\rho$) at $f = 997$ kHz. Since the emergent inductance ($L$) has the same scale of shape-dependence as the electrical resistance, $L$ and Im $\rho^{1f}$ (the first harmonic component Im $\rho$ for the ac current excitation at frequency $f$) is connected by the following relation with cross-section ($S$) and voltage-terminal distance ($d$) of device:

$$L = \frac{\text{Im}\rho^{1f} d}{2\pi f S}.$$

Therefore, Im $\rho$ can work as a good index to evaluate a material-specific EEMI. Im $\rho$ shows large signals below $T_{C1}$, which become more pronounced at lower temperatures, confirming that the emergent inductance arises from the current-induced dynamics of spin textures in this material. Note here that the EEMI is only active for the helical spin textures with $q$//$j$ // [10-10] at $T < T_{C1}$ and not for the case of $qcj$ at $T_{C1} < T < T_{C2}$. It also shows a large nonlinearity with respect to current density, like the case of Re $\rho$. What is special for the EEMI of this material is the magnitude of Im $\rho$. At $T = 10$ K, it exceeds 10 μΩ cm in some cases, and the quality factor, defined as Im $\rho$ / Re $\rho$, reaches several percent even at such a low frequency as $f = 997$ kHz. These are an order of magnitude higher than the values for previously reported emergent inductor materials with similar current density ($j_0 \sim 10^8$ A/m$^2$) and frequency (~1 kHz). In the present device size, Im $\rho = 10$ μΩ cm corresponds to $L = 14$ μH.

To investigate the emergent inductance of Tb$_5$Sb$_3$ in more detail, we have measured the current density and frequency dependence of the emergent inductance using an LCR meter. Here, another device structure of Tb$_5$Sb$_3$ was used for the two-terminal LCR measurement; the device (#2) size was 1.5 μm × 9.3 μm in cross section ($S$) and 36.4 μm long ($d$) along the [10-10] direction. Figure 4a shows the frequency dependence of the inductance $L = \text{Im} \rho d/2\pi f S$ at $T = 10$ K at several current densities. Figure 4b plots the current density dependence of the inductance at three fixed frequencies. Although qualitatively different frequency dependencies can be observed depending on the current density, all of them are attenuated rapidly with increasing $f$ above the cutoff frequency of several kHz. Here we can sort the emergent inductance response according to three regimes of current density; (1) Low current density regime ($j_0 < 2 \times 10^8$ A/m$^2$): positive inductance is observed, while the cutoff frequency remains relatively high. The inductance asymptotically approaches a constant value in the low frequency limit. (2) Intermediate current density regime ($2 \times 10^8$ A/m$^2 < j_0 < 4 \times 10^8$ A/m$^2$): A very large positive inductance value (amounting to 30-70 μH), but the cutoff frequency becomes relatively low. (3) High current density regime ($4 \times 10^8$ A/m$^2 < j_0$): A negative inductance dip appears around $f = 5$ kHz in the $f$ dependence. Figure 4c shows the frequency dependence of the inductance measured at various temperatures from $T = 10$ K to 60 K while maintaining a low current density ($j_0 = 0.73 \times 10^8$ A/m$^2$). As the temperature rises up to $T_{C1}$ (~60 K), the signal monotonically decreases in intensity without appreciable change of the cutoff frequency. In the higher-temperature magnetic phase with the $q$//$c$ conical state ($T_{C1} < T < T_{C2}$), as well as in the paramagnetic phase ($T > T_{C2}$), essentially no inductance signal, indicating that all the emergent inductance observed here is due to the ac-current induced tilting dynamics of the

disordered in-plane helimagnetic texture with a helix period of about 4 nm.

In the high current density region, the appearance of *negative* inductance as well as its highly current-nonlinear characteristic in the well-ordered helimagnetic states, *e.g.*, proper-screw and transverse conical states, has been interpreted in terms of the contribution of the pinned phason mode, namely the bound mode of the spin-rotational motion[20–22]. The large enhancement of the negative inductance is assigned to the proximity of the ac current frequency to the resonance at the phason pinning frequency[17]. Such a simple picture of the phason mode dynamics may not be applied straightforwardly to the present case of highly disordered helimagnetic textures. Nevertheless, it is reasonable to assume that the large *positive* inductance in the low and intermediate current density regions mostly originates from the tilting mode of the spin helix. Then, the question arises as to why such a qualitatively different behavior shows up between the low/intermediate current density regions and the high current density region. The previous study on $(Y, Tb)Mn_6Sn_6$ has revealed that spin fluctuations assisted by thermal excitation enhances the positive inductance which is evident only around the magnetic ordering temperature[22]. However, this is not the case in the present material; since the lower the temperature, the larger the positive inductance is. The inductance signal is almost completely eliminated near the magnetic transition temperature, and thus the main cause for the positive inductance is not due to thermal fluctuations of spin moments. One possibility is the EEMI associated with magnetic domain wall motions. In a previous study[22], a steep *positive* peak of inductance was observed at the phase boundary between helimagnetic ordered phases, suggesting that the domain wall can be a source of relatively large inductance. The disordered helimagnetic state (Fig. 1d) with a distribution of $q$-values can be viewed as the ensemble of the small-size helimagnetic ordered regions accompanied by a high density of chiral domain walls, which over the entire sample can give rise to large positive inductance signals. The transition of EEMI generation mechanism from the spin tilt-motion in the disordered helix to the spin rotational motion with the increase of the current density is likely to occur, while a quantitative description of the spin-texture dynamics is beyond our scope. However, it is worth to note again that the local spin non-collinearity in the highly disordered helimagnetic texture can generate an enhanced positive emergent inductance at low frequencies ($f < 500$ Hz) and at a moderate current density ($2$-$4 \times 10^7$ A/m$^2$), as shown in Fig. 4b.

To conclude, we have investigated the spin texture and emergent magneto-transport phenomena on single crystalline samples of $Tb_5Sb_3$. Through SANS measurements, we found that the lowest temperature magnetic phase seems to be a highly disordered helimagnetic phase with a broad distribution of helical wavenumber (a basic primitive helical unit is ~4 nm). At low temperatures below the in-plane helimagnetic transition temperature (~55 K), the charge transport under in-plane current is greatly affected by the magnetic disorder and shows a large negative current-nonlinear resistance. In accord with this feature, the low-temperature state upon the ac-current excitation shows a huge positive inductance, an order of magnitude larger than the emergent inductance values so far observed for helimagnets of comparably short (2-4 nm) helical pitches[20–22]. Such an enhancement of the emergent inductance may be ascribed to the short correlation length (~1 nm) of the spin helix in this compound, which may promote the current-drivable tilting spin motion in analogy to the spin-helix domain-wall excitations observed in other compounds[21,22]. The emergent inductance is observed to eventually change sign from positive to negative as further increasing the current density above $3 \times 10^8$ A/m$^2$, indicating the change of the current-driven spin-helix mode from tilting type to rotation type. With this finding, we establish $Tb_5Sb_3$ and related compounds with disordered helimagnetism as a promising materials platform for studying emergent electromagnetic induction phenomena. Specifically, other $Mn_5Si_3$-type magnets, which exhibit different types of helimagnetism or cluster spin glass phases[24–28], may also have the potential to exhibit non-trivial and gigantic current non-linear inductance responses. Furthermore, this research will add a value of emergent induction function to a broad family of magnetic compounds with disordered magnetic systems, which is known for advantageous functionalities. Disorder in magnetism is often tunable and can enable control of magnetic properties and be controlled by variation of magnetic field. The present study opens a future research path for possible tuning of the EEMI through control of disorder.

## Methods

### Small Angle Neutron Scattering (SANS) experiments
SANS experiments were carried out using SANS-I instrument at the Swiss Spallation Neutron Source (SINQ), Paul Scherrer Institut, Switzerland. The single-crystalline samples grown by the self-flux method[33] were attached onto an Al plate holder and loaded into a cryomagnet (base temperature, 2 K) installed at the beamline. The scattered neutrons were collected by a two-dimensional multidetector placed 2.5 m behind the sample. The neutron beam with the wavelength of 5 Å was collimated over a distance of 4.5 m before the sample, and incident nearly parallel to the *c*-axis. SANS detector measurements were carried out over a range of cryomagnet rotation angles ('rocking angles') to capture obtain the magnetic diffraction patterns on the *ab*-plane. Data obtained above $T_{C2}$ were also obtained and subtracted from the low temperature data in order to leave just the low temperature magnetic scattering. Due to the instrument construction at the high-**q** region of interest, the profile of neutron absorption across the detector was uneven. For the quantitative analysis, we only analyzed the left-hand-side of the detector which had the uniform transmission profile.

### Fabrication of device and transport measurements
To fabricate the devices for transport measurements, we cut thin plates out of the $Tb_5Sb_3$ single crystals by using the focused ion beam (FIB) technique (NB-5000, Hitachi). The thin plates were mounted on silicon substrates with patterned electrodes. We fixed the thin plates to the substrates and electrically connected them to the electrodes by using FIB-assisted tungsten-deposition. We made Au/Ti-bilayer electrode patterns by an electron-beam deposition method.

The temperature dependence of the complex resistivity was measured with use of lock-in amplifiers (LI5650, NF Corporation). We input a sine-wave current and recorded both in-phase ($ReV^{1f}$) and out-of-phase ($ImV^{1f}$) voltages with a standard four-terminal configuration. Background signals were estimated by measuring a short circuit where the terminal pads were connected by Au/Ti-bilayer electrode patterns.

The frequency dependence of the complex inductance was measured with use of LCR meter (Agilent Technologies, E4980A). We corrected the contributions from the cables and the electrodes with a standard open/short-circuit correction procedure. Here, the observed complex impedance $\widetilde{Z}(\omega)$ is the sum of frequency-independent resistance ($R$) and frequency-dependent reactance of complex inductance $[\omega \widetilde{L}(\omega) = \omega ReL(\omega) + i\omega ImL(\omega)]$. The real and imaginary components of inductance can be estimated as $ReL(\omega) = Im\widetilde{Z}(\omega)/\omega, ImL(\omega) = (Re\widetilde{Z}(\omega) - R)/\omega$.

## Data availability
All data needed to evaluate the findings of the paper are available within the paper itself. Additional data related to this paper are available from the corresponding author upon reasonable request.

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

## Acknowledgements

We would like to thank Naoto Nagaosa for enlightening discussions. This work was supported by Core Research for Evolutional Science and Technology (CREST), Japan Science and Technology Agency (JST) (Grant No. JPMJCR1874 and Grant No. JPMJCR20T01), Precursory Research for Embryonic Science and Technology (PRESTO) Japan Science and Technology Agency (JST) (Grant No. JPMJPR23Q3), Fusion Oriented Research for Disruptive Science and Technology (FOREST), Japan Science and Technology Agency (JST) (Grant No. JPMJFR2038), the Japan Society for the Promotion of Science (JSPS) KAKENHI (Grant No. JP23K19029, JP23H05431, JP23H04017, JP22K18965 and No. JP21J11830), Hirose Foundation, Mitsubishi Foundation, the special fund of Institute of Industrial Science, The University of Tokyo, The Thermal & Electric Energy Technology Foundation (TEET), Fujimori Science and Technology Foundation, Iketani Science and Technology Foundation (ISTF), TANAKA Precious Metals and the Swiss National Science Foundation (SNSF) Sinergia network "NanoSkyrmionics" (Grant No. CRSII5_171003), the SNSF Project No. 200021_188707 and an ETH Zürich Research Partnership Grant RPG 072021_07. This work is based partly on experiments performed at the Swiss spallation neutron source SINQ, Paul Scherrer Institute, Villigen, Switzerland.

## Author contributions

Y.T. conceived the project. Y. O. and A.K. grew the single crystals. A. K. fabricated the devices and performed transport measurements with the assistance from N.K. J. S. W. and V. U. carried out SANS experiments. L. P., X. Y. and K. N. took LTEM images. All authors discussed the results and commented on the manuscript.

## Competing interests

The authors declare no competing interests.
