## [Peer Review File · Communications Physics]

Reviewers' comments:

Reviewer #1 (Remarks to the Author):

The authors have investigated the temperature dependent spin structures of Tb₅Sb₃ single crystals and found two magnetic phases below 133 K and 55 K exhibiting a conical spin phase with a propagation vector along and perpendicular to the c-axis, respectively. Investigation of the AC resistivity reveals in the imaginary part of ρ , which is connected to the inductance, a strongly current and frequency dependent behavior, partly with negative, partly with enhanced positive emergent inductance. The authors attribute their experimental findings to the transition from spin-tilting motion to rotational motion upon increase of the current accompanied with the existence of domain walls due to considerable disorder of the helical spin state.

The huge emergent inductance is an order of magnitude larger than for any other known helical magnet. Moreover, the behavior differs from a similar system, (Y,Tb)Mn₆Sn₆, which has been under investigation in a previous study published by the same group. The presented data are sound, and the conclusion is comprehensible. Therefore, I recommend in principle the manuscript for publication, but prior to publication I would like to invite the authors to revise their manuscript to address the following specific concerns:

(i) For me the conical spin phase as described in the text is not so clear. The authors state that at $T < T_{C1}$ the "q vector propagates in the ab-plane as illustrated in Fig. 1c" (line 125). However, in Fig. 1c the q vector is not parallel to the ab-plane but is tilted by a certain angle. The authors should alter their presentation in order to make clear what is meant here. In addition, it would be helpful if there is an illustration of the spin-tilt mode and the phason mode as, e. g., done in Ref. [20], Fig. 1a,b.

(ii) I do not understand what is meant by the statement "Since the emergent inductance (L) has the same dimension as the electrical resistance,..." (lines 196-197). For me, the inductance has the dimension $[M \cdot L^2 \cdot T^{-2} \cdot I^{-2}]$, whereas resistance has $[M \cdot L^2 \cdot T^{-3} \cdot I^{-2}]$. The authors should give a comment, what they intend to say.

(iii) A minor point is the different typesetting of the angle "phi" in the formula in line 53 and in the describing sentence in line 59 which can easily be fixed. A similar issue is the denomination of the resistivity in the caption of Fig. 3 (line 5), where the resistivity is denoted as "r" and not as "rho".

Reviewer #2 (Remarks to the Author):

In this paper, the authors investigate emergent inductance arising from a spiral magnetic structure, employing the specific material Tb5Sb5. The results successfully separate the real and imaginary parts of time-dependent electrical resistance, leading to the successful estimation of emergent inductance from the imaginary part. Supplementary material includes real-space magnetic structures, and their consistency with reciprocal space structures is considered. The experimental procedures and results themselves are reasonable, and the work demonstrates novelty.

However, the following points (1-4) require revision:

(1) How does the primary result of this study differ from the previously reported emergent inductance phenomenon originating from spiral magnetic structures? Is the only novel aspect the presence of disordered helimagnetism? The uncontrollability of the disordered structure appears to conflict with functionality. If so, what are the advantages of using this material?

(2) Regarding the crucial result (Figure 4a), additional comments are needed on possible origins of the appearance of negative inductance over a wide range for strong currents exceeding $3.6 \times 10^8 \text{ A/m}^2$. While this phenomenon is reported as stemming from the spiral magnetic structure-induced inductance (spin motive force), possible experimental origins (both intrinsic and extrinsic) where negative inductance is observed should be enumerated, separate from theoretical proposals. Is the negativity related to the disordered structure? If so, why?

(3) In the basic formula,

$$L = \frac{\text{Im} \{ \rho_{1f} \}}{2 \pi f S},$$

which forms the basis for interpreting the experimental results, requires justification in terms of experimental data. That is, does the relationship between Fig. 3b and Fig. 4a accurately reflect this ($L \propto \text{Im} \{ \rho_{1f} \}$)?

(4) Inadequacy of fairness in the citation in the introduction: the well-known formula,

$$e_i = \frac{h}{2 \pi e} \mathbf{n} \cdot \left(\partial_i \mathbf{n} \times \partial_t \mathbf{n} \right),$$

was first given by Volovik [9], not (never!) by Refs.[4-5,8]. The result of Ref. [1] (in the References of the present manuscript) is an application of this formula to the case of moving magnetic spiral, triggered by external current (but its microscopic origin, specifically for the domain wall motion, was given by Tataro

and Kohno, PRL92, 086601 (2004)). Furthermore, the spin motive force (=emergent inductance) generated by the spiral magnetic structure is explicitly argued in Ref. [14], and the microscopic mechanism of current-driven motion of the magnetic spiral was proposed in Kishine, Ovchinnikov, Proskurin, PRB82, 064407 (2010). The originality of these earlier works should be fairly presented without any confusion.

Reviewer #3 (Remarks to the Author):

Referee report on "Enhanced emergent electromagnetic inductance in Tb5Sb3 due to highly disordered helimagnetism," by A. Kitaori, et al.

This paper investigates the emergent electromagnetic induction (EEMI) phenomena observed in Tb5Sb3, focusing on its highly disordered helimagnetic spin texture. At low temperatures, below the in-plane helimagnetic ordering temperature (~ 55 K), the charge transport under in-plane current is significantly influenced by the magnetic disorder, leading to a highly nonlinear I-V behavior. However, this state, when probed with an ac current, exhibits a large positive inductance, which is notably higher than previously reported values for helimagnets with similar short helical pitches (2-4 nm). The enhancement of emergent inductance is attributed to the short correlation length (~ 1 nm) of the spin helix in this compound, which likely facilitates the current-drivable tilting spin motion. The study also reveals that the emergent inductance changes sign from positive to negative as the current density increases beyond 3×10^8 A/m², indicating a transition from tilting to rotational current-driven spin-helix modes. These results position Tb5Sb3 and similar compounds with disordered helimagnetism as promising materials for exploring EEMI phenomena, suggesting potential for novel applications in miniaturized electronic components.

This is an interesting paper with the theme that disorder can sometimes improve the functionality of devices, even though the behavior becomes more difficult to understand.

The introduction is nicely done, providing sufficient background and a judicious choice of references. The experimental data are of high quality. The SANS data are sufficient to show that this system has a short-range helical spin structure, and using Lorentz TEM it was confirmed that it has a single-Q helical spin structure (with multiple domains). Additionally, changes in the stripe pattern with temperature indicates the instability of the magnetic structure. However, it would be good if the authors could make a plot of the peak intensity as a function of the polar angle with varying temperatures.

As Tb₅Sb₃ is a Yoshimori-type helimagnet, chiral magnetic domains are expected. It would be useful if the authors could include some discussion on how these domains might affect their results. Are the authors confident their sample has only one domain? How do they know? Will the contributions from the different domains tend to cancel?

One other question involves the use of W to make contacts to the sample. Can the authors comment on the contact resistance they achieve with these contacts? Are the contacts Ohmic? Why was W chosen over, say, Pt?

If the authors address these minor concerns I enthusiastically recommend this paper for publication.

Referee Report for the Manuscript titled "Emergent Inductance from Spiral Magnetic Structures in Tb_5Sb_5 "

In this paper, the authors investigate emergent inductance arising from a spiral magnetic structure, employing the specific material Tb_5Sb_5 . The results successfully separate the real and imaginary parts of time-dependent electrical resistance, leading to the successful estimation of emergent inductance from the imaginary part. Supplementary material includes real-space magnetic structures, and their consistency with reciprocal space structures is considered. The experimental procedures and results themselves are reasonable, and the work demonstrates novelty.

However, the following points require revision:

1. How does the primary result of this study differ from the previously reported emergent inductance phenomenon originating from spiral magnetic structures? Is the only novel aspect the presence of disordered helimagnetism? The *uncontrollability* of the disordered structure appears to conflict with functionality. If so, what are the advantages of using this material?
2. Regarding the crucial result (Figure 4a), additional comments are needed on possible origins of the appearance of negative inductance over a wide range for strong currents exceeding $3.6 \times 10^8 \text{ A/m}^2$. While this phenomenon is reported as stemming from the spiral magnetic structure-induced inductance (spin motive force), possible experimental origins (both intrinsic and extrinsic) where negative inductance is observed should be enumerated, separate from theoretical proposals. Is the negativity related to the disordered structure? If so, why?
3. In the basic formula,

$$L = \frac{\text{Im}\rho^{1f}d}{2\pi fS},$$

which forms the basis for interpreting the experimental results, requires justification in terms of experimental data. That is, does the relationship between Fig. 3b and Fig. 4a accurately reflect this ($fL \propto \text{Im}\rho^{1f}$)?

4. Inadequacy of fairness in the citation in the introduction: the well-known formula,

$$e_i = \frac{h}{2\pi e} \mathbf{n} \cdot (\partial_i \mathbf{n} \times \partial_t \mathbf{n}),$$

was first given by Volovik [9], not (never!) by Refs.[4-5,8]. The result of Ref. [1] (in the References of the present manuscript) is an application of this formula to the case of moving magnetic spiral, triggered by external current (but its microscopic origin, specifically for the domain wall motion, was given by Tatara and Kohno, PRL92, 086601 (2004)). Furthermore, the spin motive force (=emergent inductance) generated by the spiral magnetic structure is explicitly argued in Ref. [14], and the microscopic mechanism of current-driven motion of the magnetic spiral was proposed in Kishine, Ovchinnikov, Proskurin, PRB82, 064407 (2010). The originality of these earlier works should be fairly presented without any confusion.

Reviewers' comments:

Reviewer #1 (Remarks to the Author):

The authors have investigated the temperature dependent spin structures of Tb₅Sb₃ single crystals and found two magnetic phases below 133 K and 55 K exhibiting a conical spin phase with a propagation vector along and perpendicular to the *c*-axis, respectively. Investigation of the AC resistivity reveals in the imaginary part of ρ , which is connected to the inductance, a strongly current and frequency dependent behavior, partly with negative, partly with enhanced positive emergent inductance. The authors attribute their experimental findings to the transition from spin-tilting motion to rotational motion upon increase of the current accompanied with the existence of domain walls due to considerable disorder of the helical spin state.

The huge emergent inductance is an order of magnitude larger than for any other known helical magnet. Moreover, the behavior differs from a similar system, (Y,Tb)Mn₆Sn₆, which has been under investigation in a previous study published by the same group. The presented data are sound, and the conclusion is comprehensible. Therefore, I recommend in principle the manuscript for publication, but prior to publication I would like to invite the authors to revise their manuscript to address the following specific concerns:

(i) For me the conical spin phase as described in the text is not so clear. The authors state that at $T < T_{C1}$ the “*q* vector propagates in the *ab*-plane as illustrated in Fig. 1c” (line 125). However, in Fig. 1c the *q* vector is not parallel to the *ab*-plane but is tilted by a certain angle. The authors should alter their presentation in order to make clear what is meant here. In addition, it would be helpful if there is an illustration of the spin-tilt mode and the phason mode as, e. g., done in Ref. [20], Fig. 1a,b.

[Reply]

First of all, the authors thank the reviewer for the constructive comments, all of which are useful to make the present paper more persuasive.

The magnetic structure of Tb₅Sb₃ at the lowest temperatures is superposition of mainly three components. The first is a cycloidal magnetic structure that propagates in the *ab*-plane, the second is a ferromagnetic component in the *ab*-plane, and the third is a ferromagnetic component parallel to the *c*-axis. Judging from these three components, a conical phase, in which the cone is tilted, is realized. Here, while the cone axis is tilted from the both of *ab*-plane and *c*-axis, the *q* vector remains within the *ab* -plane. We have revised the Fig. 1d to make easier to understand that the propagation vector is perpendicular to the *c*-axis.

Also, following the reviewer’s suggestion, we have added schematic illustrations of the spin-tilt mode and the phason mode as Fig. 1a.

(ii) I do not understand what is meant by the statement “Since the emergent inductance (*L*) has the same dimension as the electrical resistance,...” (lines 196-197). For me, the inductance has the dimension $[M \cdot L^2 \cdot T^{-2} \cdot I^{-2}]$, whereas

resistance has $[M \cdot L^2 \cdot T^{-3} \cdot I^{-2}]$. The authors should give a comment, what they intend to say.

[Reply]

We thank the reviewer for pointing out the misleading expressions. Unlike classical inductance, emergent inductance is inversely proportional to the cross-sectional area and proportional to the inter-electrode distance. To clarify our intent to express this fact, we have changed the expression as follows;

<lines 197-198>

Since the emergent inductance (L) has the same dimension as the electrical resistance,

→ Since the emergent inductance (L) has the same scale of shape -dependence as the electrical resistance

(iii) A minor point is the different typesetting of the angle “phi” in the formula in line 53 and in the describing sentence in line 59 which can easily be fixed. A similar issue is the denomination of the resistivity in the caption of Fig. 3 (line 5), where the resistivity is denoted as “r” and not as “ρ”.

[Reply]

Thank you for pointing out the typo. We have fixed it.

Reviewer #2 (Remarks to the Author):

In this paper, the authors investigate emergent inductance arising from a spiral magnetic structure, employing the specific material Tb5Sb5. The results successfully separate the real and imaginary parts of time-dependent electrical resistance, leading to the successful estimation of emergent inductance from the imaginary part. Supplementary material includes real-space magnetic structures, and their consistency with reciprocal space structures is considered. The experimental procedures and results themselves are reasonable, and the work demonstrates novelty.

However, the following points (1-4) require revision:

(1) How does the primary result of this study differ from the previously reported emergent inductance phenomenon originating from spiral magnetic structures? Is the only novel aspect the presence of disordered helimagnetism? The uncontrollability of the disordered structure appears to conflict with functionality. If so, what are the advantages of using this material?

[Reply]

We thank the reviewer for his/her deep understanding and appreciation of the objective and accomplishments of the present study.

As the reviewer commented, the uniqueness of this system lies in its disordered helimagnetism. Through this study, it was revealed that disorder system can dramatically enhance emergent inductance. Such disordered spin textures have been studied as cluster spin-glasses, and several material systems are known. We hope that this research will be able to bring about novel functionality from a new perspective to a broader range of magnetic materials system. We

have added a sentence to further emphasize this point at lines 285-290. In this context, we also highlight that disorder in magnetism is known to offer advantageous functionalities in other areas, such as using spin glasses for secure memory devices. Additionally, disorder is often tunable, and can enable control of magnetic properties, e.g. support topological effects and be controlled by variation of magnetic field. The present study opens a future research path for possible tuning of the EEMI through control of disorder.

(2) Regarding the crucial result (Figure 4a), additional comments are needed on possible origins of the appearance of negative inductance over a wide range for strong currents exceeding $3.6 \times 10^8 \text{ A/m}^2$. While this phenomenon is reported as stemming from the spiral magnetic structure-induced inductance (spin motive force), possible experimental origins (both intrinsic and extrinsic) where negative inductance is observed should be enumerated, separate from theoretical proposals. Is the negativity related to the disordered structure? If so, why?

[Reply]

Negative inductance-like behavior itself has been observed in all the other materials investigated so far, therefore it cannot be immediately associated with magnetic disorder. Well-ordered helical magnets such as YMn_6Sn_6 and $\text{Gd}_3\text{Ru}_4\text{Al}_{12}$ host negative inductance (Ref. 20-22). In those well-ordered helimagnets, the emergent inductance due to the phason mode should show a sign change at extrinsic pinning frequency (ω_p), and become negative at a lower frequency region around ω_p as described in Ref. 22. We consider that the widespread predominance of *positive* emergent inductance is a rather novel behavior characteristic of this material that exhibits the disordered helical magnetism. This may be correlated with the fact that such a magnetic structure like that presented by Tb_5Sb_3 , can be as interpreted in terms of a structure with dense domain walls that is unlikely to host the current-induced well-defined phason-like excitation, as already argued in the main text.

(3) In the basic formula,

$$L = \frac{\text{Im} \rho^{1f}}{2\pi f S},$$

which forms the basis for interpreting the experimental results, requires justification in terms of experimental data. That is, does the relationship between Fig. 3b and Fig. 4a accurately reflect this ($L \propto \text{Im} \rho^{1f}$)?

[Reply]

As for the data of Fig. 3b and Fig. 4a, even though they were measured on different devices, the proportional relationship is almost satisfied. As an example, at $j = 7.3 \times 10^7 \text{ A/m}^2$, $f \sim 1 \text{ kHz}$, $T = 10 \text{ K}$, the imaginary part of the resistivity of sample #2 (used in Fig. 4) can be calculated as $\text{Im} \rho^{1f} = 3.4 \mu\Omega \text{ cm}$. Considering that 1 kHz is near to the cutoff frequency, this value is close to the value $5.1 \mu\Omega \text{ cm}$ of sample #1 (used in Fig. 3) under similar conditions.

(4) Inadequacy of fairness in the citation in the introduction: the well-known formula,

[

$$e_i = \frac{\hbar}{2\pi e} \mathbf{n} \cdot \nabla \left(\frac{\partial \mathbf{n}}{\partial t} \times \mathbf{n} \right),$$

was first given by Volovik [9], not (never!) by Refs.[4-5,8]. The result of Ref. [1] (in the References of the present manuscript) is an application of this formula to the case of moving magnetic spiral, triggered by external current (but its microscopic origin, specifically for the domain wall motion, was given by Tataru and Kohno, PRL92, 086601 (2004)). Furthermore, the spin motive force (=emergent inductance) generated by the spiral magnetic structure is explicitly argued in Ref. [14], and the microscopic mechanism of current-driven motion of the magnetic spiral was proposed in Kishine, Ovchinnikov, Proskurin, PRB82, 064407 (2010). The originality of these earlier works should be fairly presented without any confusion.

[Reply]

We thank the reviewer for this important comment. Accurately identifying the contributions of these earlier studies is of great benefit to the readers and greatly enhances the integrity of our manuscript. Following the reviewer's advice, we provided appropriate citations and added references to their work in the introduction at lines 52-58.

Reviewer #3 (Remarks to the Author):

Referee report on "Enhanced emergent electromagnetic inductance in Tb5Sb3 due to highly disordered helimagnetism," by A. Kitaori, et al.

This paper investigates the emergent electromagnetic induction (EEMI) phenomena observed in Tb5Sb3, focusing on its highly disordered helimagnetic spin texture. At low temperatures, below the in-plane helimagnetic ordering temperature (~55 K), the charge transport under in-plane current is significantly influenced by the magnetic disorder, leading to a highly nonlinear I-V behavior. However, this state, when probed with an ac current, exhibits a large positive inductance, which is notably higher than previously reported values for helimagnets with similar short helical pitches (2-4 nm). The enhancement of emergent inductance is attributed to the short correlation length (~1nm) of the spin helix in this compound, which likely facilitates the current-drivable tilting spin motion. The study also reveals that the emergent inductance changes sign from positive to negative as the current density increases beyond 3×10^8 A/m², indicating a transition from tilting to rotational current-driven spin-helix modes. These results position Tb5Sb3 and similar compounds with disordered helimagnetism as promising materials for exploring EEMI phenomena, suggesting potential for novel applications in miniaturized electronic components.

This is an interesting paper with the theme that disorder can sometimes improve the functionality of devices, even though the behavior becomes more difficult to understand.

The introduction is nicely done, providing sufficient background and a judicious choice of references. The experimental data are of high quality. The SANS data are sufficient to show that this system has a short-range helical spin structure, and using Lorentz TEM it was confirmed that it has a single-Q helical spin structure (with multiple

domains). Additionally, changes in the stripe pattern with temperature indicates the instability of the magnetic structure. However, it would be good if the authors could make a plot of the peak intensity as a function of the polar angle with varying temperatures.

[Reply]

We would like to thank the reviewers for his/her deep understanding and evaluation of the purpose and results of this study.

We agree that a plot of peak intensity as a function of polar angle is particularly meaningful at temperatures just below the transition temperature (and more specifically, it's more appropriate based on bulk SANS data than on LTEM using thin-plate samples). Unfortunately, at this stage, it is difficult to say that the quality of the S/N ratio is quantitatively satisfactory as shown in the below graph (Fig. R1), due to the instrument construction at the high-q region of interest. For the quantitative analysis we only analyzed the left-hand-side of the detector which had the uniform transmission profile. Regarding this, we would like to accumulate more comprehensive data in future research.

Fig. R1. Angle-dependence of SANS intensity. Due to uneven shielding by the instrument, the height of each peak has become significantly different even though they are at crystallographically equivalent positions.

As Tb5Sb3 is a Yoshimori-type helimagnet, chiral magnetic domains are expected. It would be useful if the authors could include some discussion on how these domains might affect their results. Are the authors confident their sample has only one domain? How do they know? Will the contributions from the different domains tend to cancel?

[Reply]

As the reviewer mentioned, chiral magnetic domains exist in centrosymmetric helimagnets, and we also recognize that there are multiple domains in the single sample. However, since the emergent inductance does not depend on whether the chirality is right-handed or left-handed, the cancellation stemming from the chiral multi-domains does not occur. Nevertheless, as mentioned in Ref. 22, an enhancement of emergent inductance is observed at phase boundaries, so it is possible that domain walls (including those caused by chiral magnetic domains) have the effect of enhancing emergent inductance. We have made slight changes in wording in the revised manuscript to convey this point more clearly.

<lines 256-259>

The disordered helimagnetic state (Fig. 1d) with a distribution of q-values can be viewed as the ensemble of the small-size helimagnetic ordered regions accompanied by a high density of chiral domain walls, which over the entire sample can give rise to large positive inductance signals.

One other question involves the use of W to make contacts to the sample. Can the authors comment on the contact resistance they achieve with these contacts? Are the contacts Ohmic? Why was W chosen over, say, Pt?

If the authors address these minor concerns I enthusiastically recommend this paper for publication.

[Reply]

We thank the reviewer for his/her recommendation.

In this device using W deposition, the electrical resistance of the contact is as small as 0.1-0.2 Ω , and we are confident that the essential electrical signal of the sample is reflected. In our environment, Pt tends to have a slightly higher contact resistance than W; therefore, unless there are some concerns, we preferentially use tungsten as the electrode.

Summary of changes made:

-We have added schematic illustrations of the spin-tilt mode and the phason mode (Fig. 1a) and revised Fig. 1d.

-We have added a comment on the previous research and several references (lines 52-58)

-We have added a comment on the cluster spin glass materials (lines 285-290)

-We have added a comment on the SANS setup and analysis (lines 304-306)

-We have changed several expressions (lines 197-198, 258-259)

REVIEWERS' COMMENTS:

Reviewer #1 (Remarks to the Author):

The authors have revised their manuscript according to the reviewers' comments. They have added some illustrations and have changed some text passages which improved the quality of the paper and made it easier to understand. Therefore, I recommend the manuscript in its present form for publication.

Reviewer #2 (Remarks to the Author):

After reviewing the revised manuscript, I find that satisfactory revisions have been made in response to the review comments. Therefore, I agree to proceed with publication.

Reviewer #3 (Remarks to the Author):

I have carefully read the authors' reply to the Referees and the revised manuscript. In my view, the manuscript has been improved and is suitable for publication.